# FAMESUMM: Investigating and Improving Faithfulness of Medical Summarization

**Nan Zhang[1], Yusen Zhang[1], Wu Guo[2], Prasenjit Mitra[1,3], Rui Zhang[1]**

[1]The Pennsylvania State University
[2]Children's Hospital Affiliated of Zhengzhou University
[3]L3S Research Center
{njz5124,yfz5488,pmitra,rmz5227}@psu.edu,   guoscho@126.com

## Abstract

Summaries of medical text shall be faithful by being consistent and factual with source inputs, which is an important but understudied topic for safety and efficiency in healthcare. In this paper, we investigate and improve faithfulness in summarization on a broad range of medical summarization tasks. Our investigation reveals that current summarization models often produce unfaithful outputs for medical input text. We then introduce FAMESUMM, a framework to improve faithfulness by fine-tuning pre-trained language models based on medical knowledge. FAMESUMM performs contrastive learning on designed sets of faithful and unfaithful summaries, and it incorporates medical terms and their contexts to encourage faithful generation of medical terms. We conduct comprehensive experiments on three datasets in two languages: health question and radiology report summarization datasets in English, and a patient-doctor dialogue dataset in Chinese. Results demonstrate that FAMESUMM is flexible and effective by delivering consistent improvements over mainstream language models such as BART, T5, mT5, and PEGASUS, yielding state-of-the-art performances on metrics for faithfulness and general quality. Human evaluation by doctors also shows that FAMESUMM generates more faithful outputs. Our code is available at https://github.com/psunlpgroup/FaMeSumm.

## 1 Introduction

Summarizing medical text is a key step towards improving the efficiency in healthcare (Liu et al., 2019; Krishna et al., 2020), including applications in summarizing medical dialogues (Joshi et al., 2020), clinical notes (Kanwal and Rizzo, 2022), health questions (He et al., 2021), and radiology reports (Dai et al., 2021).

An important but understudied issue of medical text summarization is faithfulness, as defined in recent literature (Maynez et al., 2020; Huang et al., 2021): a summary is unfaithful if it contains intrinsic error (the fact that contradicts the source) or extrinsic error (the fact that cannot be directly inferred from the source text), while a faithful summary should be free from both. Unfaithful summaries will pose significant healthcare risks by misleading patients and medical providers. However, to the best of our knowledge, only a few papers investigate the faithfulness of medical summarization by providing systematic categorization of errors for medical domain (Otmakhova et al., 2022; Adams et al., 2022). Additionally, recent investigation and faithfulness improvement (Zhang et al., 2020b; Alambo et al., 2022) works are limited to certain types of medical text such as radiology reports.

Moreover, as many summarization approaches are based on language models that are pretrained on general domain text, they are inadequate to generate faithful summaries due to the lack of medical domain knowledge. For example, although He et al. (2021) and Zhang et al. (2021) leveraged pretrained models such as T5 (Raffel et al., 2019) and BART (Lewis et al., 2020), their performance is suboptimal because they did not incorporate medical knowledge into their models.

In this work, we aim to both investigate and improve the faithfulness of medical summarization. We first provide a taxonomy of faithfulness error types appearing in medical summaries and analyze several competitive baselines. Our investigation in Table 1 shows that current summarization models make a significant amount of faithfulness errors. Then, we introduce FAMESUMM to improve **Fa**thfulness for **Me**dical **Summ**arization. FAMESUMM is a general-purpose framework applicable to various language models on many medical summarization tasks. It adopts two objectives that finetune pre-trained language models to explicitly model faithfulness and medical knowledge. The first one uses contrastive learning (Khosla et al., 2020; Cao and Wang, 2021). FAMESUMM adopts

| Category | Description | Example | Percentage |
|---|---|---|---|
| | | Intrinsic Errors | |
| Entity Relationship | Expressing wrong relations between two entities or actions. | **Source**: 医生: ...化验支原体抗体，阳性用红霉素治疗。
**Prediction**: 输液的,可以去医院用红霉素检查... | 5.25% |
| Entity | Generating wrong entities. | **Source**: 患者:本人男17岁，身高176cm体重61.3kg...
**Prediction**: 患者身高1717cm体重61.3kg... | 3.00% |
| Negation | Ignoring or adding negation words. | **Source**: 患者:尘肺在潜伏期可以检查出来嘛（男，19岁）
医生:你好。不能检查出来。
**Prediction**: 尘肺在潜伏期可以检查出来。 | 3.50% |
| Question | Misinterpreting a question into a statement or vice versa. | **Source**: 患者:意思是如果没控制好，也会使宝宝脑瘫了?
**Prediction**: 如果没控制好,也会使宝宝脑瘫了。 | 0.75% |
| | | Extrinsic Errors | |
| Template | Generating statement that follows the patterns (template) of training samples. | **Prediction**: No evidence of acute cardiopulmonary process.*
**Gold**: Minimal blunting of the right costophrenic angle, suggesting a small pleural effusion. No focal infiltrate. | 18.00% |
| Extraneous Fact | Add additional facts that are not in the source. | **Prediction**: No evidence of active or latent tuberculosis.
**Gold**: Unremarkable radiographic examination of the chest. No radiographic evidence of tuberculosis. | 14.50% |
| Low Specificity | Not specific enough to describe the situation. | **Prediction**: What are the treatments for vomiting?
**Gold**: What can I give my 8-month-old for constant vomiting? | 2.00% |

Table 1: A taxonomy of faithfulness error categories of the baseline models. The translation of Chinese texts is provided in Appendix A. Words marked in waves are the errors and those marked in underlines are the evidence from the source or reference summary to infer the errors. * This sentence is a common pattern in the training set and thus is a template.

much simpler heuristics (as straightforward as rule-based copying and manipulating source texts) than other contrastive learning baselines and achieve state-of-the-art performances. The second objective learns medical knowledge by modeling medical terms and their contexts in the loss function. We show that directly modeling context tokens of medical terms is an effective design for faithfulness improvement, which offers enrichment for existing approaches to learning medical knowledge (Joshi et al., 2020; Michalopoulos et al., 2022).

We apply FAMESUMM to a variety of backbone pretrained language models on three datasets including health question summarization, radiology report summarization, and medical dialogue summarization. We compare FAMESUMM with baselines including backbone models, faithfulness-based models, state-of-the-art medical summarizers, and GPT-3 as an example of large language models (LLMs). FAMESUMM generates 16% more faithful summaries than GPT-3 based on doctors' evaluation, and it provides consistent score improvements over baselines according to automatic metrics. Unlike recent methods (Adams et al., 2022; Chen et al., 2022) that train additional models for faithfulness, as a cost-efficient method, FAMESUMM demonstrates the possibility of maximizing faithfulness by designing simple contrastive sets and incorporating medical knowl-

edge. Our contributions are: (1) We investigate existing summarization models for medical text to reveal their faithfulness issues in the medical domain; (2) We propose the FAMESUMM framework with two fine-tuning strategies to improve faithfulness; (3) We conduct a comprehensive set of experiments to demonstrate the effectiveness of FAMESUMM by improving mainstream language models and achieving state-of-the-art performances on several benchmarks.

## 2 Investigating Faithfulness of Medical Summarization

Comprehensive error analysis can improve the understanding of the severity of faithfulness issues, and help researchers better design a model for improving faithfulness. However, little research has studied the faithfulness issues in medical summarization. Therefore, we provide a systematic taxonomy of faithfulness errors. Specifically, we first randomly pick 100 samples from HQS, RRS, and MDS datasets (Section 4.1), respectively, forming 300 samples in total. We use fine-tuned transformer models and the pointer-generator network as baselines to generate summaries for each sample, and we manually check and categorize the faithfulness errors of the summaries. Inspired by Maynez et al. (2020), we first classify the faithfulness errors

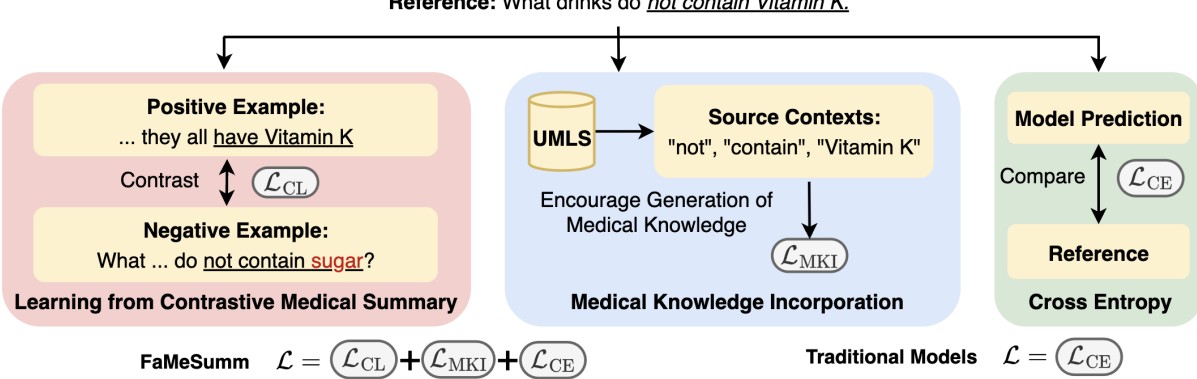

Figure 1: Diagram of FAMESUMM architecture with an example reference summary. The underlined part in the reference contains a medical term ("Vitamin K") and its context ("do not contain") that are modeled by FAMESUMM.

found in the 300 samples into intrinsic errors and extrinsic errors. Then, we further cluster the error cases in a finer granularity based on their main characteristics: (1) The intrinsic error includes entity error, entity relationship error, negation error, question error, and modifier error, and (2) the extrinsic error contains template error, low specificity error, and extraneous fact error. Complementing the error types in Pagnoni et al. (2021) with new types that are specific to the medical domain (*e.g.*, low specificity error), Table 1 shows examples of each error type produced by our baselines along with their popularity. As shown in the table, 47.00% of the samples have faithfulness errors, demonstrating that current summarization models make a significant amount of faithfulness errors. Among all errors, entity relationship/template errors (5.25%, 18.00%) are the most common intrinsic/extrinsic errors, respectively.

## 3 Improving Faithfulness of Medical Summarization

We propose FAMESUMM, a faithfulness optimization framework guided by medical domain knowledge for pretrained language models. As shown in Figure 1, the first objective $\mathcal{L}_{\text{CL}}$ uses contrastive learning (**CL**) over a set of curated faithful and unfaithful summaries created by perturbing medical entities and concepts (Section 3.1); the second objective $\mathcal{L}_{\text{MKI}}$ explicitly performs medical knowledge incorporation (**MKI**) and encourages generating accurate medical terms from source texts (Section 3.2). These two strategies complement each other during the fine-tuning stage in addition to the cross-entropy loss $\mathcal{L}_{\text{CE}}$ (Section 3.3). FAMESUMM is flexible because it is widely applicable to various backbone pretrained language models.

### 3.1 Learning from Contrastive Summaries

For our first objective, we design positive and negative sets of summaries guided by medical entities, and we use contrastive learning to train models to distinguish faithful and unfaithful summaries. Medical terms are defined in Section 4.4.

We build a positive and a negative set for every training instance to implement contrastive learning (Cao and Wang, 2021). A positive set ($P$) contains faithful summaries of the source text while a negative set ($N$) contains unfaithful and suspected unfaithful summaries (references with unmentioned medical terms).

**Designing Contrastive Medical Summaries**    We use two steps to build the positive set.

- **Reference summary with faithfulness validation.** Reference summaries have been found to contain unfaithful contents (Maynez et al., 2020; Dhingra et al., 2019), and thus we perform a faithfulness validation before adding them to the positive set. Specifically, we apply a simple heuristic algorithm that a reference summary will be in the positive set if all its medical concepts show up in the corresponding source. Although a reference summary with unmentioned medical terms does not necessarily make itself unfaithful, we do notice the faithfulness issues between source and reference in both our dataset and existing literature. Since having unmentioned medical terms is a good indicator of extraneous information, we put a reference with unmentioned medical terms into the negative set[1].
- **Data augmentation by sentence extraction.** We further augment positive summaries by extract-

---

[1]We experiment with placing all references into the positive sets but notice suboptimal faithfulness performance as described in Section 5.

ing sentences from the source. Depending on the placement of the reference summary from the previous strategy, we extract one or several sentences (or utterances in dialogue) from the source as positive example(s) to meet the minimum size requirement of the positive set. We extract content in the following order of priority: **(1)** extract a sentence that contains medical terms showing up in both source and reference if there is one, **(2)** extract the longest sentence (or the last utterance in dialogue), **(3)** extract the first utterance (usually a question posted by a patient) if the data is dialogue, and **(4)** apply back-translation by translating the sentence from (1), (2), or (3) to German and then back to English as a paraphrase.

The negative set is constructed by manipulating sentences from source text by the following:

- **Reference summary failing faithfulness validation.** Based on the verification of reference in positive set construction, a reference summary with any unmentioned medical terms will be a negative example.

- **Replacing and appending medical concepts.** When a reference summary contains a medical term showing up in the source, we replace that medical term with a new one. We also add a new medical concept at the start or the end of the reference summary. These newly introduced terms break the faithfulness of the original reference summary because they are randomly selected and do not appear in the source or the reference.

- **Changing attribute value.** If a reference summary contains numerical attributes, we change the number to another random value, *e.g.*, "5 doses" will be changed to "6 doses".

- **Entity swap.** To break the relationships among entities in a reference summary, we first apply named entity recognition and swap all recognized entities in the original summaries.

- **Logic inversion.** We invert the logic of a reference summary in our Chinese dataset so that its affirmative sentence changes to a negative one or vice versa. To achieve this, we first select a pair of positive and negative unigrams (Appendix H). Whenever we see a reference that has the positive or the negative unigram in this pair, we then change it to invert the logic.

 Given the positive and negative sets, the contrastive learning loss function $\mathcal{L}_{\text{CL}}$ is:

$$\mathcal{L}_{\text{CL}} = -\frac{1}{\binom{|P|}{2}} \sum_{\substack{y_i, y_j \in P \\ y_i \neq y_j}} \log \frac{\exp(\cos(\boldsymbol{h}_i, \boldsymbol{h}_j)/\tau)}{\sum_{\substack{y_k \in P \cup N \\ y_i \neq y_k}} \exp(\cos(\boldsymbol{h}_i, \boldsymbol{h}_k)/\tau)}$$

where $\boldsymbol{h}_i$, $\boldsymbol{h}_j$, $\boldsymbol{h}_k$ are representations for summaries $y_i$, $y_j$, and $y_k$ (we use outputs produced by the last layer of the decoder from a language model, and the outputs are averaged over all sequence tokens). $\cos(\cdot, \cdot)$ computes cosine similarity, and $\tau$ is temperature. By minimizing $\mathcal{L}_{\text{CL}}$, we finetune the model to generate summary representations to maximize the discrepancy between contrastive sets and develop a preference for more faithful outputs.

### 3.2 Incorporating Medical Knowledge

Our second objective aims to incorporate medical knowledge in the reference to generate more faithful summaries. To this end, we design a loss term $\mathcal{L}_{\text{MKI}}$ to encourage models to maximize the likelihood of the medical terms in reference by increasing their generation probability.

We first identify all medical terms in reference summaries, since this type of term is one of the most important carriers of medical information. For each medical concept, we further consider its context including (1) two tokens preceding the medical term and (2) any negative unigram (Appendix H) in the reference. Then, we maintain a vector $b_m$ with the same length as the vocabulary size to record the frequency of the context tokens and medical terms. For example, if a token of interest appears two times in a reference summary, its corresponding value in $b_m$ is 2. Let $p$ denote model prediction scores of all vocabularies, *i.e.*, logits of all vocabulary tokens averaged across all reference tokens before the softmax layer. Since $b_m$ and $p$ have the same dimension, $\mathcal{L}_{\text{MKI}}$ is computed as the dot product of $b_m$ and $p$:

$$\mathcal{L}_{\text{MKI}} = -b_m \cdot p \tag{1}$$

In this way, $\mathcal{L}_{\text{MKI}}$ encourages our model to maximize the prediction scores of the medical terms and their context words during generation to improve summary faithfulness.

### 3.3 Overall Fine-tuning Objective

As described above, $\mathcal{L}_{\text{CL}}$ and $\mathcal{L}_{\text{MKI}}$ incorporate medical knowledge to promote faithfulness. Furthermore, we add the cross-entropy loss $\mathcal{L}_{\text{CE}}$ to obtain our final training objective:

$$\mathcal{L} = \lambda_{\text{CL}} \cdot \mathcal{L}_{\text{CL}} + \lambda_{\text{MKI}} \cdot \mathcal{L}_{\text{MKI}} + \mathcal{L}_{\text{CE}} \tag{2}$$

where $\lambda_{\text{CL}}$ and $\lambda_{\text{MKI}}$ serve as weights. Note that we encourage all the medical terms in reference summaries due to $\mathcal{L}_{\text{MKI}}$ while discouraging unmentioned medical terms due to $\mathcal{L}_{\text{CL}}$.

| Summarization Task | Source | Language | Split (# Instances) | Disease Distribution | | | | | | |
|---|---|---|---|---|---|---|---|---|---|---|
| | | | | Heart | Liver | Brain | Kidney | Respiration | Stomach | Others |
| Health Question (HQS) | NLM | English | 1000/50/100 | 44 | 17 | 51 | 19 | 37 | 28 | 970 |
| Radiology Report (RRS) | Hospital | English | 91544/4000/600 | 34461 | 1069 | 4277 | 303 | 76433 | 6505 | 11195 |
| Medical Dialogue (MDS) | Telemedicine | Chinese | 1346/288/288 | 63 | 106 | 162 | 102 | 150 | 124 | 1322 |

Table 2: Statistics of three datasets for evaluation. We show the number of train/dev/test examples in the split.

## 4 Experiment Setup

### 4.1 Datasets

To demonstrate the applicability of our approach as a general-purpose summarizer, we use three different datasets in Table 2 including Health Question Summarization (HQS) from the MEDIQA 2021 shared task 1 (Ben Abacha et al., 2021; Ben Abacha and Demner-Fushman, 2019), Radiology Report Summarization (RRS) from the MEDIQA 2021 shared task 3 (Johnson et al., 2019a,b; Demner-Fushman et al., 2016), and Medical Dialogue Summarization (MDS) (Song et al., 2020).

### 4.2 Evaluation Metrics

We adopt a holistic evaluation scheme in aspects of faithfulness, general quality, and human evaluation.
**Faithfulness Metrics** To measure faithfulness, we report three metrics from existing literature based on question answering, medical entity, or textual entailment: QuestEval (Scialom et al., 2021), FaR (Shing et al., 2021), and SummaC (Laban et al., 2022). We also use Concept F1 (C F1) to measure the coverage of medical concepts in reference summaries (Joshi et al., 2020).
**General Quality** We report ROUGE-1/2/L (Lin, 2004) and BERTScore (Zhang et al., 2019) to assess the general summarization quality.
**Human Evaluation by Doctors** We conduct a human evaluation by medical experts to judge coherence and faithfulness. Six doctors are recruited: three of them evaluate 100 test instances in HQS while the other three are on 100 test instances in MDS. Doctors are asked to identify incoherent and unfaithful summaries produced by all candidate models. The order of summaries generated by models is randomized and anonymous to doctors. When all doctors complete their evaluation, they discuss how to resolve major disagreements while still holding their distinctive opinions towards individual examples. We then count the number of faithful and coherent summaries through a vote of three doctors. Two types of voting are reported: (1) **consensus**: a summary is considered faithful or coherent if at least two doctors think so, and (2)

**strict**: a summary is faithful or coherent when all three doctors agree.

### 4.3 Baselines

Five types of baselines are considered:
**Backbone pretrained language model** We fine-tune Pegasus-large (Zhang et al., 2020a), BART-large (Lewis et al., 2020), BioBART-large (Yuan et al., 2022), T5-base (Raffel et al., 2019), and mT5-small (Xue et al., 2021) as our baselines.
**Different variants of FAMESUMM** To sort out the factors that affect performance, we present two FAMESUMM variants on HQS including training with contrastive learning only and training with positive sets that accept all references. When using mT5-small on MDS, we also conduct an ablation study to showcase the effectiveness of individual fine-tuning objectives.
**Faithfulness-based model** Since contrastive learning is a major part of FAMESUMM, CLIFF (Cao and Wang, 2021) and QFCL (Zhang et al., 2022) are the closest baselines we find. CLIFF is shown to be effective for faithfulness improvement on news-related datasets while QFCL is built specifically for medical question summarization. QFCL is different from CLIFF in terms of negative set construction method.
**Abstractive medical summarizer** We select Joshi et al. (2020) as the medical dialogue summarizer that fits MDS. Other related work (Krishna et al., 2021; Zhang et al., 2021) target special types of medical dialogue (*e.g.*, extremely long dialogue), which is not the focus of this paper.
**LLM** To make model comparison fair, we do not consider zero-shot summarization from LLMs. We fine-tune GPT-3 (Brown et al., 2020) on 10 random training instances of HQS as a baseline.

### 4.4 Implementation Details

FAMESUMM relies on the identification of medical terms. For English datasets, we use Unified Medical Language System (Bodenreider, 2004) to identify them. For MDS in Chinese, we recruit human annotators to tag them as described in Appendix B.

| Model | Faithfulness | | | | General Quality | | | |
|---|---|---|---|---|---|---|---|---|
| | QuestEval | FaR | SummaC | C F1 | R1 | R2 | RL | BERTScore |
| PEGASUS (He et al., 2021) | 0.3069 | 0.3188 | 0.4279 | 26.24 | 30.19 | 11.93 | 28.52 | 0.7427 |
| + CLIFF (Cao and Wang, 2021) | 0.3145 | 0.3217 | 0.4225 | 30.18 | 29.92 | 11.40 | 27.67 | 0.7401 |
| + QFCL (Zhang et al., 2022) | 0.3193 | 0.3042 | 0.4322 | 28.00 | 29.27 | 11.01 | 27.32 | 0.7396 |
| + FAMESUMM (CL only) | 0.3157 | 0.3222 | 0.4355 | 30.20 | 29.31 | 10.82 | 27.39 | 0.7352 |
| + FAMESUMM (All ref in $P$)† | 0.3105 | 0.3267 | 0.4138 | 28.90 | 31.14 | 12.29 | 29.09 | 0.7489 |
| + FAMESUMM | 0.3134 | 0.3492 | 0.4401 | 30.92 | 30.84 | 12.19 | 28.99 | 0.7456 |
| BART (He et al., 2021) | 0.3120 | 0.3300 | 0.4062 | 30.02 | 31.51 | 10.57 | 29.84 | 0.7512 |
| + FAMESUMM | 0.3127 | 0.3233 | 0.4640 | 40.42 | 31.76 | 11.71 | 29.64 | 0.7491 |
| T5 (He et al., 2021) | 0.3126 | 0.3237 | 0.4125 | 27.38 | 30.11 | 11.40 | 27.54 | 0.7460 |
| + FAMESUMM | 0.3186 | 0.3467 | 0.4342 | 32.79 | 30.19 | 11.00 | 27.91 | 0.7470 |
| BioBART (Yuan et al., 2022) | 0.3116 | 0.3487 | 0.4810 | 30.15 | 32.38 | 11.91 | 29.51 | 0.7486 |
| + FAMESUMM | 0.3239 | 0.3595 | 0.4740 | 33.33 | 32.99 | 12.57 | 30.56 | 0.7544 |

Table 3: Results for HQS. We rerun baselines to compute all metrics. † places all references into positive sets.

| Model | Faithfulness | | | | General Quality | | | |
|---|---|---|---|---|---|---|---|---|
| | QuestEval | FaR | SummaC | C F1 | R1 | R2 | RL | BERTScore |
| | RRS-Indiana | | | | | | | |
| PEGASUS (Dai et al., 2021) | 0.2413 | 0.0682 | 0.2350 | 46.12 | 44.35 | 29.67 | 43.87 | 0.7999 |
| + FAMESUMM | 0.2441 | 0.0741 | 0.2769 | 47.37 | 46.69 | 33.13 | 46.15 | 0.8111 |
| | RRS-Stanford | | | | | | | |
| PEGASUS (Dai et al., 2021) | 0.2892 | 0.2146 | 0.4907 | 40.74 | 40.97 | 26.37 | 38.45 | 0.7755 |
| + FAMESUMM | 0.2884 | 0.2324 | 0.4931 | 42.86 | 41.21 | 26.90 | 38.86 | 0.7768 |

Table 4: Results for RRS. We rerun baselines to compute all metrics. We show two splits separately.

| Model | FaR | C F1 | RL | BERTScore |
|---|---|---|---|---|
| Joshi et al. (2020) | 0.0619 | 32.53 | 19.99 | 0.6937 |
| mT5* | 0.3784 | 36.96 | 32.43 | 0.7505 |
| +CL* | 0.4040 | 37.85 | 32.73 | 0.7518 |
| +MKI* | 0.4072 | 39.30 | 32.27 | 0.7492 |
| +FAMESUMM * | 0.4177 | 38.57 | 33.05 | 0.7520 |

Table 5: Results for MDS. For models with *, we report their scores that are averaged over 3 random seeds (seed values 42, 0, and 1) due to unstable training dynamics of transformer-based models (Mosbach et al., 2021). Scores based on a single seed (single-run scores) are reported in Appendix I.

More implementation details are in Appendix C. Due to differences on language and dataset structure, we customize the contrastive set construction for each dataset, and the details are in Appendix F and G. Hyperparameters of each model are in Appendix J.

## 5 Results and Analysis

All performance scores in this section are evaluated on the test set of each dataset. Our results

and analysis aim to answer the following research questions:

- RQ 1: How can FAMESUMM improve faithfulness of backbone models (5.1)?
- RQ 2: What are the effects of the two loss functions in FAMESUMM (5.2 and 5.5)?
- RQ 3: How does FAMESUMM compare with other types of baselines including faithful summarization models (5.1), state-of-the-art medical text summarization (5.3), and LLMs (5.4)?
- RQ 4: Is performing faithfulness validation on reference summaries better than accepting all of them into the positive sets (5.1)?
- RQ 5: How do doctors judge the faithfulness of medical summarization and its correlation with current faithfulness metrics (5.4)?

### 5.1 Overall Results

Our overall results for three datasets are presented in Table 3, 4, and 5.

**Improvements over Backbone Models** As discussed in Section 4.2, the effects of learning from

| Dataset | Model | R1 | R2 | RL |
|---------|-------|-----|-----|-----|
| HQS | Sänger et al. (2021) | 33.40 | 15.99 | 31.49 |
| | He et al. (2021) | 35.14 | 16.08 | 31.31 |
| | FAMESUMM + Ensemble | 35.35 | 14.86 | 33.06 |
| RRS-Indiana | Mahajan et al. (2021) | 67.72 | 58.81 | 66.57 |
| | Dai et al. (2021) | 68.34 | 59.56 | 67.17 |
| | FAMESUMM + Ensemble | 68.30 | 59.68 | 67.46 |
| RRS-Stanford | Mahajan et al. (2021) | 38.84 | 22.84 | 36.11 |
| | Dai et al. (2021) | 43.12 | 27.69 | 40.14 |
| | FAMESUMM + PEGASUS | 43.34 | 28.11 | 40.46 |
| MDS | Joshi et al. (2020) | 22.81 | 8.34 | 19.99 |
| | mT5 | 34.20 | 19.95 | 32.43 |
| | FAMESUMM + mT5 | 34.92 | 20.56 | 33.05 |

Table 6: Ensembling results of FAMESUMM. We report ROUGE compared with previous state-of-the-art on each benchmark.

contrastive medical summaries are demonstrated by improved automatic faithfulness measures. The effects of medical knowledge incorporation are mainly demonstrated through C F1. With only three exceptions on QuestEval, FaR, and SummaC metrics (minimal decrease), FAMESUMM provides consistent improvements over the corresponding backbone models on all automatic faithfulness metrics. Therefore, we show that the proposed fine-tuning strategies are effective at learning medical knowledge and generating more faithful outputs. ROUGE and BERTScore are not compromised with FAMESUMM, since most of them are higher when compared against corresponding backbone models.

**Advantage of Faithfulness Validation on References** Comparing the last two rows of PEGASUS-based models in Table 3, we see that faithfulness validation on reference summaries enables FAMESUMM to dominate all faithfulness metrics. ROUGE and BERTScore are slightly lower after adding this validation, since we disvalue many references by placing them into the negative sets. We include this validation as a step of building contrastive sets, since the decrease is trivial and this paper mainly deals with improving faithfulness.

**Improvements over Faithful Summarization Baselines** Compared with CLIFF or QFCL in Table 3, FAMESUMM (CL only) offers a consistent improvement on all faithfulness metrics despite a small decrease on QuestEval, which demonstrates the advantage of our contrastive learning method. There are two reasons: (1) both CLIFF and QFCL do not enforce any validation of references, and this is a flawed design based on Adams et al. (2022) and the discussion above, and (2) our CL considers much more comprehensive error types than CLIFF

| Model | Consensus | | Strict | | Fleiss' kappa |
|-------|-----------|-----|--------|-----|---------------|
| | # C | # F | # C | # F | |
| | | | HQS | | |
| GPT-3 | 100 | 60 | 100 | 37 | 0.4675 |
| PEGASUS | 100 | 71 | 100 | 54 | 0.4295 |
| + CLIFF | 100 | 70 | 100 | 59 | 0.6508 |
| + FAMESUMM [†] | 100 | 73 | 100 | 60 | 0.4976 |
| + FAMESUMM | 100 | 76 | 100 | 66 | 0.4697 |
| | | | MDS | | |
| Joshi et al. (2020) | 100 | 22 | 54 | 1 | 0.7316 |
| mT5 | 100 | 82 | 100 | 77 | 0.7479 |
| + MKI | 100 | 84 | 100 | 81 | 0.7946 |
| + FAMESUMM | 100 | 89 | 100 | 83 | 0.6351 |

Table 7: Human evaluation results. [†] places all references into positive sets. "# C": Coherent. "# F": Faithful. We report Fleiss' kappa on faithfulness.

| Model | Dataset | Spearman's $\rho$ | | |
|-------|---------|-------------------|-----|---------|
| | | QuestEval | FaR | SummaC |
| FAMESUMM +PEGASUS | HQS | 0.067 | 0.171 | 0.312 |

Table 8: Correlation analysis between human evaluation and three faithfulness metrics.

and QFCL based on our observation in Section 2. The advantage of our CL demonstrates the utility of designing more diverse and accurate contrastive sets to learn a decision boundary. In Table 5, FAMESUMM and all mT5-based models are better than Joshi et al. (2020) due to the success of pre-trained transformer models.

**Complementary Roles of CL and MKI** The performance gap between FAMESUMM and CLIFF (or QFCL) becomes much more significant after we add MKI, which is shown through the last row of PEGASUS-based models in Table 3. Most faithfulness metrics are positively impacted when we train with the complete version of FAMESUMM (FaR experiences the greatest increase), and general quality metrics of FAMESUMM are also significantly higher than CLIFF and QFCL. Although CL and MKI have different focuses, their complementary roles enable performance boost across a variety of metrics.

### 5.2 Ablation Study of Two Loss Functions

As discussed above, both fine-tuning strategies in FAMESUMM contribute to the overall faithfulness improvement. In Table 5, we iteratively add a fine-tuning strategy to mT5. As a result, FaR score keeps increasing when we add more strategies. The overall quality as measured by RL and BERTScore

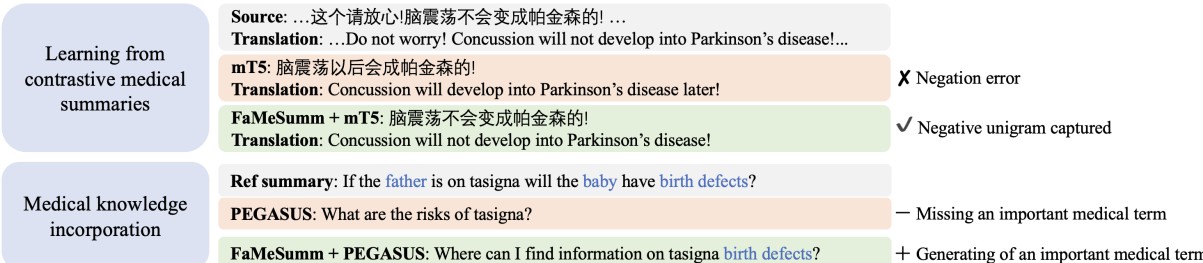

Figure 2: Two examples (from MDS and HQS datasets) to show the effect of FAMESUMM finetuned on mT5 and PEGASUS. We mark the medical terms in blue in the second example.

is also the highest with both strategies. `C F1` is slightly lower on the last row of Table 5, indicating that our pipeline is learning a trade-off between our two strategies.

We can see that MKI is most effective on entity-based faithfulness metrics. It yields the highest `C F1` and the second highest `FaR`. Relative to MKI, CL improves faithfulness more comprehensively in a mixture of question answering, entailment, and entity aspects. CL has better `RL` and `BERTScore` than MKI, which indicates that improving faithfulness in comprehensive ascepts has the potential of improving general summarization quality. CL also has better scores than plain mT5.

### 5.3 Ensembling for Comparison with State-of-the-art Medical Summarization

Since the best papers on HQS and RRS adopt model ensembling, we perform ensembling over multiple FAMESUMM results to compare with those works. Table 6 displays this comparison, which demonstrates that our proposed fine-tuning strategies can yield state-of-the-art performance. In general, we could achieve better performance than the best paper on each benchmark of MEDIQA with much fewer base models or ad-hoc techniques. For example, as the best paper on HQS, He et al. (2021) utilized ensembling based on four models along with training on the validation set and error correction for misspellings. Our ensembled outputs (with FAMESUMM) surpass the best score on `R1` and `RL` without using the validation set or any error correction techniques. More ensembling details are provided in Appendix E. On MDS, we see a constant `ROUGE` increase from Joshi et al. (2020) to FAMESUMM. One reason is due to the superiority of pre-trained language models over pointer-generator networks in Joshi et al. (2020). Besides that, FAMESUMM yields better scores than plain mT5 because of FAMESUMM's capability of generating domain-specific and more faithful summaries.

Although this paper mainly deals with faithfulness concerns in healthcare, we show that improving faithfulness can also yield better ROUGE scores due to the fact that FAMESUMM output stays consistent with the source text.

### 5.4 Human Evaluation

We report our human evaluation results in Table 7. FAMESUMM outputs the most faithful summaries on both HQS and MDS, which reconfirms its strongest capability of improving faithfulness. Echoing the analysis in Section 5.1, FAMESUMM without faithfulness validation has suboptimal performance. Joshi et al. (2020) suffers from incoherent summaries and only has 1 faithful summary under the strict standard (doctors will not mark a summary as faithful if it is incoherent). We find moderate to substantial inter-annotator agreement for faithfulness judgments. Fleiss' kappas on MDS are higher than those on HQS (except for CLIFF), both of which are above $0.4$.

**Compare with GPT-3** GPT-3 only has 37 faithful outputs under the strict protocol. FAMESUMM generates 16% more faithful summaries than GPT-3 based on consensus protocol. Doctors find that GPT-3 generates more extrinsic errors than other models (*e.g.*, extraneous fact error in Table 1). Its linkage to external knowledge allows its outputs to be more diverse, but it also yields more unrelated or even incorrect information.

**Correlation Analysis** We show the correlation between human evaluation and the three faithfulness metrics in Table 8. All correlation scores are small. `SummaC` has the strongest correlation while `QuestEval` has very weak correlation. Our findings of the low correlation are echoed in Wang et al. (2023) and call for future research efforts on faithfulness metrics for medical summarization. This also reflects our motivation for conducting doctors' evaluations.

## 5.5 Case Study

We present two examples in Figure 2 to show the effectiveness of the two loss functions in FAMESUMM. In the contrastive learning example, FAMESUMM copied a key sentence from the source while capturing an important negative unigram ("not"). However, plain mT5 ignored it and thus generated a negation error. FAMESUMM is more likely to capture this kind of negative unigram than its baseline because we applied **logic inversion** when we built the negative sets for the MDS dataset.

In the second example, FAMESUMM successfully generates an important medical term "birth defects", while the baseline failed. This term makes its summary more specific and consistent with the reference. Incorporating medical knowledge, FAMESUMM is more likely to generate medical concepts than its baselines to improve faithfulness.

## 6 Related Work

**Medical Summarization** There are different medical summarization datasets on medical dialogues (Joshi et al., 2020; Zeng et al., 2020; Krishna et al., 2021; Yim and Yetisgen, 2021; Navarro et al., 2022; Zhou et al., 2021) and multi-documents (DeYoung et al., 2021; Katsimpras and Paliouras, 2022). Some of them are private. MEDIQA 2021 shared tasks (Ben Abacha et al., 2021) tackle three summarization tasks: consumer health question summarization (HQS), multi-answer summarization (MAS), and radiology report summarization (RRS).

As for medical summarization models, Joshi et al. (2020) and Enarvi et al. (2020) constructed pointer-generator networks, so they did not benefit from the success of pre-trained transformer models. Zhang et al. (2021) fine-tuned BART and developed a multistage approach to handle long conversations. The length of conversations is not the concern of our work. The focus of Krishna et al. (2021) is to generate SOAP notes (Subjective information, Objective observation, Assessments made by doctors, and Plan for future care), and their work aims to have summaries divided into at most 15 sections. By contrast, FAMESUMM serves as a general-purpose fine-tuning strategy for abstractive summarization in healthcare with wide applicability.

**Faithfulness and Factuality in Abstractive Summarization** Recent work has proposed several factual evaluation metrics (Fabbri et al., 2021; Scialom et al., 2021; Wang et al., 2020; Fabbri et al., 2022; Laban et al., 2022; Shing et al., 2021). Cao and Wang (2021) utilized contrastive learning (Khosla et al., 2020) with a design of contrastive sets in the news domain for faithfulness enhancement. Several recent efforts (Goyal et al., 2022; Liu et al., 2022; Tam et al., 2022) investigated summarization factuality of LLMs such as GPT-3 (Brown et al., 2020) and BLOOM (Scao et al., 2022), yet they only used news-related datasets. Few papers (Zhang et al., 2020b; Alambo et al., 2022) that directly modeled faithfulness in healthcare (by adding inductive bias informed by faithfulness through model optimization) focused on single specific tasks such as radiology report summarization. By contrast, we show that it is important to design domain-specific contrastive sets and offer a general-purpose medical summarizer.

## 7 Conclusion

In this paper, we investigate faithfulness issues on abstractive medical summarization in current summarization models. We propose FAMESUMM consisting of two fine-tuning objectives that can be applied to mainstream pretrained language models across many datasets. Our results based on automatic metrics and doctor evaluation show that our method mitigates the faithfulness concern in medical summarization and delivers state-of-the-art performances on several benchmarks.

## Limitations

Since FAMESUMM involves learning from contrastive summaries, it sets a relatively high requirement on GPU memory, which is a common drawback of existing contrastive learning framework. For any dataset used for training, adding the contrastive learning component of FAMESUMM will demand more memory than training without it. As a result, the training batch size of FAMESUMM may be set to a small value. For example, fine-tuning `Pegasus-large` with FAMESUMM on RRS-Indiana takes about 48GB memory using a training batch size of 2. Note that FAMESUMM is more efficient than many recent works such as Chen et al. (2022) in terms of memory consumption and/or other training requirements as described in Section 1, and we never need to go beyond batch size

of 4 for all experiments on FAMESUMM. Our code supports multi-GPU training to allow users to try larger batch sizes.

## Ethics Statement

Improving the faithfulness of abstractive summaries is an active research area. Although methods proposed in this work are shown to be effective at reducing unfaithful errors in Section 5, completely eliminating these errors has not yet been reached. Since unfaithful errors may still exist in our model outputs, we emphasize that FAMESUMM is designed to be used under supervision from at least one medical practitioner. This medical practitioner needs to check the faithfulness of any summaries generated from FAMESUMM before providing them to patients in order to avoid risks. We used detailed statistics as well as a case study in Section 5 to showcase the performance of FAMESUMM in terms of faithfulness.

All medical datasets used in this work were de-identified before we accessed them.

## Acknowledgements

We thank Tianyang Zhao, Xiangyu Dong, Yilun Zhao, Yiyang Feng, Fangxu Yu, Yunxiang Li, Xueqing Zhang, and Wei Chen for their significant assistance on data annotation. This research was partially funded by the Federal Ministry of Education and Research (BMBF), Germany under the project LeibnizKILabor with grant No. 01DD20003.

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

## A Translation of Chinese Examples for The Taxonomy of Faithfulness Errors

In Table 9, we provide English translation of the Chinese texts in Table 1.

## B Details of Constructing Medical Dialogue Summarization Dataset

We follow Song et al. (2020) to construct our medical dialogue summarization dataset. During web crawling, we find that the URLs of all the dialogues on *Chunyu-Doctor* start with a common part[2]. Most of the dialogues have summaries (*type-A summaries*) that are concatenations of original utterances from the doctors, and only a small group of them contains summaries (*type-B summaries*) written by the doctors that may have novel words. We set the type-B summary as the reference summary and only collect the dialogues that contain type-B summaries. A type-B summary is always preceded by an identifier called *"based on this inquiry, the doctor updates the summary and suggestion:"*.

We look for URLs that start with the common part described above, and we only download their HTML pages if their source dialogues contain the identifier of the type-B summary. Due to constant changes of the URL of each dialogue on the platform, we download its source HTML immediately after we find it. Then, we scrape dialogue and summary from every HTML document collected and discard the dialogues that have images or audio inputs. We also throw away those that contain only one utterance, because they are technically not conversations between two characters. Finally, some

---

[2] https://www.chunyuyisheng.com/pc/qa/

| Category | Example (English Translation) |
|---|---|
| Entity Relationship | **Source**: Doctor: ...to test for mycoplasma antibodies and treat with erythromycin if the result is positive.
**Prediction**: receive IV therapy, you can get checked for erythromycin at the hospital... |
| Entity | **Source**: Patient: I am a 17-year-old male, with a height of 176cm and a weight of 61.3kg...
**Prediction**: The patient with a height of 1717cm and a weight of 61.3kg... |
| Negation | **Source**: Patient: Can pneumoconiosis be detected during its latency period? (Male, 19 years old)
Doctor: Hello. It cannot be detected.
**Prediction**: Pneumoconiosis can be detected during its latency period. |
| Question | **Source**: Patient: Does it mean that if it's not well controlled, it can also lead to cerebral palsy in the baby?
**Prediction**: If it's not well controlled, it can also lead to cerebral palsy in the baby. |

Table 9: English translation of Chinese examples in Table 1. Words marked in waves are the errors and those marked in underlines are the evidence from the source to infer the errors.

patients ask follow-up questions after conversations are summarized. We truncate these conversations to ensure they only contain the parts before the follow-up. Following these steps, we obtain 2251 dialogues with reference summaries. Our collected data is anonymous.

After an exploratory analysis, we notice 2 things to be completed for data cleaning. The first thing is about the specificity of the reference summary. Some reference summaries written by doctors are not directly related to their source dialogue, so we treat them as nonspecific templates. For example, in a conversation about heart disease, a summary like "please follow my account to ask further questions" are not directly related to the conversation. We need to filter out these nonspecific summaries before training our model. Another thing is to identify the medical concepts in the reference summaries, because we utilize these terms for fine-tuning FAMESUMM. Any terms that refer to disease names, treatment methods, drug names, and biomedical vocabularies are considered as medical concepts in this dataset. To maximize medical term coverage, we recruit annotators who are native speakers of Chinese to manually tag medical terms, because existing glossaries of Chinese medical terms do not fully cover those in our dataset. For instance, although Unified Medical Language System (Bodenreider, 2004) and THU Open Chinese Lexicon (Han et al., 2016) own large collections of Chinese medical concepts, many drug names from our dialogues such as "Yinaoning" (a Chinese patent drug for the growth of brain) are not found in this collection. Annotators are asked to evaluate the specificity of each reference summary and manually tag medical terms. Every dialogue comes with 3 annotation questions:

1. Does the reference summary serve as a non-

specific template to its source conversation?

2. Are there any medical terms found in the summary but not in the dialogue?

3. Are there any medical terms found in both summary and dialogue?

After the completion of annotation, we discard the dialogues that have nonspecific templates as summaries and consider the rest as our dataset for dialogue summarization. Every dialogue in this dataset comes with two sets of medical terms that correspond to annotation questions 2 and 3. We apply 70/15/15 random split on the collected data to construct our dialogue summarization dataset.

## C  More Implementation Details

FAMESUMM relies on medical terms, and the definition of medical terms or concepts depends on the different ways we use to identify them. For example, if the tool we use to tag them is Unified Medical Language System (Bodenreider, 2004), any named entity that has a CUI (Concept Unique Identifier) or is an alias of another CUI-term is considered as a medical term. We also recruited human annotators for the Chinese dialogue to do medical term identification due to the lack of promising tools. We do not align the difference of this definition, because our contributions are based on provided medical terms.

We perform grid search on the validation set of each dataset for hyperparameters tuning. For RRS (both Indiana and Stanford) and Chinese dialogue, we tune hyperparameters on their validation sets based on ROUGE scores. Because the validation data of RRS is a combined set that comes from two different sources (the data distributions of training and combined validation sets are similar but different), we train models of both RRS benchmarks

using the same training set but a different validation set. Specifically, we use "validation Indiana" for the Indiana benchmark and use the combined validation set of RRS ("validation MIMIC" and "validation Indiana") for Stanford, since Stanford does not come with its own validation data and we hope a greater data variety would help in this case.

Hyperparameter tuning is more challenging on HQS, since its training, development, and test sets are all different in terms of summarization style (He et al., 2021). Thus, we compute: (1) `Rouge-2` between model output and source text of the validation set, and (2) string length of the source text in validation divided by the string length of the output. The weights of these two values are 1 and 0.5 respectively. We use their weighted sum to tune hyperparameters for training and decoding.

## D  More Example Outputs from FAMESUMM and Baselines

In Figure 3, we show two more examples of faithfulness improvement brought by FAMESUMM in addition to the case study in Section 5.5. For the two examples in HQS, we copy the source text (a health question posted by a patient) and the outputs from CLIFF, QFCL, GPT-3, and FAMESUMM. In the first example, both CLIFF and QFCL misinterpret the source text and make faithfulness errors. In the second example, GPT-3 makes an extraneous fact error by mentioning "a skin infection caused by staphylococcus bacteria" that is never asked in the source. FAMESUMM shows strong capabilities of protecting information integrity in both examples.

## E  Model Ensembling for HQS and RRS

We provide more ensembling details here. On HQS benchmark, following the previous state-of-the-art (He et al., 2021), we rerank four system outputs (FAMESUMM fine-tuned with BioBART, T5, PE-GASUS, and BART) based on three features: fidelity, consensus, and wellformedness. Computing a weighted sum of the three features, we pick the best summary from the four outputs for every test instance. Since each of our single systems has better performance as shown in Table 3, our ensembling result achieves the new state-of-the-art.

On RRS-Indiana, the previous state-of-the-art (Dai et al., 2021) fine-tuned 16 PEGASUS models (with different seeds) on the union of training and validation data and performed ensembling based on mutual similarity scores among any two system

outputs. In contrast, we rerank the generated summaries from 7 FAMESUMM (fine-tuned based on PEGASUS with different hyperparameters) outputs based on the observations from us and Dai et al. (2021).

On RRS-Stanford, the previous state-of-the-art (Dai et al., 2021) fine-tuned 16 PEGASUS (with different seeds) on the training set with output normalization. We reach better performance by just using one FAMESUMM output. This FAMESUMM model is the same model as the last row of Table 4, except that length penalty parameter is additionally tuned on the validation set in order to encourage our model to generate longer output.

## F  Customization of Positive Sets for Each Dataset

For HQS dataset, we apply rule (1), (2), and (4) for sentence extraction. If the reference summary of a training instance is placed into the positive set, either (1) or (2) will be used in order to get one more positive example. Otherwise, (1) or (2) will be used to get the first positive example, and (4) will be used to augment it to get the second. For the experiment that places all references into the positive sets, only (4) is utilized to augment a reference summary to obtain the second positive example.

For RRS dataset, we use the same way to build positive sets as HQS.

For MDS dataset, we apply rule (1), (2), and (3) for sentence extraction. All three rules are executed to ensure we end up with at least two positive examples for each training instance. Rule (4) is not used, since we do not find a comparable machine translation tool for Chinese.

## G  Customization of Negative Sets for Each Dataset

For HQS dataset, we use all the sentence manipulation rules except logic inversion, since we find that negation error is more popular in our Chinese MDS dataset. For the experiment that places all references into the positive sets, no reference summaries will be placed into the negative set, so "reference summary failing faithfulness validation" will also not be used.

Since RRS dataset is the largest one we use, we choose not to use "changing attribute value", entity swap, and logic inversion to reduce training cost.

**Example 1**

**Source**: covid-19 how long to quarantine after being positive how long are you contagious if i tested positive for covid-19. how long before i can safely return to work after a positive covid 19 test

**CLIFF + PEGASUS**: What is the incubation period for covid-19?

**QFCL + PEGASUS**: What are the treatments for covid-19?

**GPT-3**: How long are you contagious if I tested positive for Covid-19

**FaMeSumm + PEGASUS**: How long before you can safely return to work after a positive covid 19 test?

**Example 2**

**Source**: outbreak of red raise sores on back shoulder and arms from shoulder to wrist with raised bulbs of liquid. treatment and maintenance ?

**CLIFF + PEGASUS**: What are the treatments for raised bulbs of liquid?

**QFCL + PEGASUS**: What are the treatments for raised bulbs of liquid?

**GPT-3**: What are the treatments for a skin infection caused by staphylococcus bacteria

**FaMeSumm + PEGASUS**: What are the treatments for raised bulbs of liquid?

Figure 3: Two more examples of improvement brought by FAMESUMM on HQS. We mark the faithfulness errors in red.

| Model | FaR | C F1 | RL | BERTScore |
|---|---|---|---|---|
| Joshi et al. (2020) | 0.0619 | 32.53 | 19.99 | 0.6937 |
| mT5 | 0.3973 | 37.26 | 32.80 | 0.7512 |
| +CL | 0.4133 | 37.50 | 32.99 | 0.7534 |
| +MKI | 0.4041 | 40.00 | 32.26 | 0.7497 |
| +FAMESUMM | 0.4264 | 38.24 | 33.28 | 0.7540 |

Table 10: Results for MDS based on a single seed.

For MDS dataset, we use all the sentence manipulation rules except entity swap.

## H Positive and Negative Unigrams Used by Each Dataset

As a step towards building negative sets for our Chinese dataset (MDS), we select "可以" and "不可以" as the pair of positive and negative unigrams to invert the logic of reference summaries. Their English translations are "can" and "cannot" respectively.

As a step towards incorporating medical knowledge for our English datasets (HQS and RRS), we select "no", "nope", "doesn't", "don't", and "not" as the negative unigrams to model medical concepts. As for our Chinese dataset (MDS), we select "不", "没有", "无", "没", and "非" as the negative unigrams. All these Chinese negative unigrams mean "no" in English.

## I Single-run Scores

Due to relatively unstable training dynamics, we do not want to report "lucky" or "unlucky" scores of models based on mT5. Therefore, in Table 5, these models are fine-tuned three times with seed values 42, 0, and 1. Note that all the models on the held-out validation set present the same trend as in Table 5 based on a single run (seed value 42), including Joshi et al. (2020). We report all the single-run scores on the test set of MDS in Table 10. It is clear to see that the trend is the same as in Table 5 (e.g., MKI has the largest C F1, and FAMESUMM has the best scores on all other metrics), which reinforces our analysis.

## J Hyperparameters

**PEGASUS on HQS**. The key hyperparameters are learning rate, warmup steps, and training batch size. Their values are 0.00005, 700, and 8 respectively.

**CLIFF + PEGASUS on HQS**. The key hyperparameters are learning rate, $\lambda_{CL}$, warmup steps, and training batch size. Their values are 0.00003, 1.0, 600, and 4 respectively.

**QFCL + PEGASUS on HQS**. The key hyperparameters are learning rate, $\lambda_{CL}$, warmup steps, and training batch size. Their values are 0.000025, 1.0, 600, and 4 respectively.

**FAMESUMM (CL only) + PEGASUS on HQS**. The key hyperparameters are learning rate, $\lambda_{CL}$, warmup steps, and training batch size. Their values

are 0.00003, 1.0, 600, and 4 respectively.

**FAMESUMM (All ref in P) + PEGASUS on HQS**. The key hyperparameters are learning rate, $\lambda_{CL}$, $\lambda_{MKI}$, warmup steps, and training batch size. Their values are 0.00004, 1.0, 0.001, 600, and 4 respectively.

**FAMESUMM + PEGASUS on HQS**. The key hyperparameters are learning rate, $\lambda_{CL}$, $\lambda_{MKI}$, warmup steps, and training batch size. Their values are 0.00003, 1.0, 0.001, 600, and 4 respectively.

**BART on HQS**. The key hyperparameters are learning rate, warmup steps, and training batch size. Their values are 0.000045, 1000, and 2 respectively.

**FAMESUMM + BART on HQS**. The key hyperparameters are learning rate, $\lambda_{CL}$, $\lambda_{MKI}$, warmup steps, and training batch size. Their values are 0.00003, 1.0, 0.0011, 1000, and 2 respectively.

**T5 on HQS**. The key hyperparameters are learning rate, warmup steps, and training batch size. Their values are 0.0004, 1700, and 2 respectively.

**FAMESUMM + T5 on HQS**. The key hyperparameters are learning rate, $\lambda_{CL}$, $\lambda_{MKI}$, warmup steps, and training batch size. Their values are 0.00055, 1.0, 0.0013, 1500, and 2 respectively.

**BioBART on HQS**. The key hyperparameters are learning rate, warmup steps, and training batch size. Their values are 0.00008, 0, and 2 respectively.

**FAMESUMM + BioBART on HQS**. The key hyperparameters are learning rate, $\lambda_{CL}$, $\lambda_{MKI}$, warmup steps, and training batch size. Their values are 0.00003, 0.95, 0.0011, 1000, and 2 respectively.

**PEGASUS on RRS-Indiana**. The key hyperparameters are learning rate, warmup steps, and training batch size. Their values are 0.00006, 0, and 4 respectively.

**FAMESUMM + PEGASUS on RRS-Indiana**. The key hyperparameters are learning rate, $\lambda_{CL}$, $\lambda_{MKI}$, warmup steps, and training batch size.

Their values are 0.00006, 2.0, 0.0014, 0, and 2 respectively.

**PEGASUS on RRS-Stanford**. The key hyperparameters are learning rate, warmup steps, and training batch size. Their values are 0.00006, 1000, and 4 respectively.

**FAMESUMM + PEGASUS on RRS-Stanford**. The key hyperparameters are learning rate, $\lambda_{CL}$, $\lambda_{MKI}$, warmup steps, and training batch size. Their values are 0.00004, 0.8, 0.0014, 600, and 4 respectively.

**Joshi et al. (2020) on MDS**. The key hyperparameters are learning rate, $\lambda_m$ (for magnitude of medical concept modeling), $\lambda_n$ (for magnitude of negation modeling), $\delta$ (for controlling the strength of generation probability), dimension of hidden states, and dimension of input embeddings. Their values are 0.15, 1.0, 0.1, 1.0, 256, 128 respectively.

**mT5 on MDS**. The key hyperparameters are learning rate, warmup steps, and training batch size. Their values are 0.00055, 1000, and 4 respectively.

**CL + mT5 on MDS**. The key hyperparameters are learning rate, $\lambda_{CL}$, warmup steps, and training batch size. Their values are 0.0005, 1.0, 1200, and 4 respectively.

**MKI + mT5 on MDS**. The key hyperparameters are learning rate, $\lambda_{MKI}$, warmup steps, and training batch size. Their values are 0.0005, 0.0014, 1200, and 4 respectively.

**FAMESUMM + mT5 on MDS**. The key hyperparameters are learning rate, $\lambda_{CL}$, $\lambda_{MKI}$, warmup steps, and training batch size. Their values are 0.0005, 1.0, 0.001, 1000, and 4 respectively.