# OpenReview forum: "FaMeSumm: Investigating and Improving Faithfulness of Medical Summarization"
_EMNLP/2023/Conference — EMNLP 2023 Main_

### Official Review · Reviewer_UEQF · 2023-08-04

**Soundness:** 4

**Excitement:**

4: Strong: This paper deepens the understanding of some phenomenon or lowers the barriers to an existing research direction.

**Paper Topic And Main Contributions:**

This paper describes the augmentation of recent summarization systems with methods to improve their fidelity in respecting the meaning of the original documents.
It starts by studying the level of faithfulness errors in current systems and then propose several means to improve them (creating more contrastive summaries, incorporating medical knowledge and cross entropy loss). Previous systems are the fine-tuned using the proposed learning objective.
Results on several datasets show the effectiveness of the proposed fine-tuning methods.

**Reasons To Accept:**

The paper propose a proven effective solution to a real problem in current summarization systems. The solution is well described and well evaluated. Results are good and adequately discussed.

**Reasons To Reject:**

Section 3.1  is not clear enough. In particular, the paragraph on "Data augmentation by sentence extraction" is modifying  reference summaries by adding new sentences from the source. It is not clear how this change in summary length will impact the capacity of summarizers.

**Reproducibility:**

3: Could reproduce the results with some difficulty. The settings of parameters are underspecified or subjectively determined; the training/evaluation data are not widely available.

**Reviewer Confidence:**

3: Pretty sure, but there's a chance I missed something. Although I have a good feel for this area in general, I did not carefully check the paper's details, e.g., the math, experimental design, or novelty.

**Typos Grammar Style And Presentation Improvements:**

The paper should include translations of examples in Table 1 to help non-Chinese reading readers to understand the errors.

---

> ### Author Rebuttal · Authors · 2023-08-29
>
> We thank the reviewer for the helpful feedback!
>
> Weakness1: Regarding the change in summary length of positive sets, first of all, we do notice a significant length difference between reference summaries and positive examples for some datasets. For example, in HQS dataset, its reference summaries used for training have an average number of 10.1 tokens per instance, but its positive examples have an average number of 33.2 tokens per instance. As demonstrated in Table 3, 4, 5, and 7, this difference has a positive effect on faithfulness and summarization quality, because combining reference summaries with these longer positive examples form a diverse training data which helps FaMeSumm to better learn a decision boundary about generating tokens.
>
> We will add translation for examples in Table 1 and a clarification about the change in summary length of positive sets in our final version.

---

### Official Review · Reviewer_fbJn · 2023-08-04

**Soundness:** 3

**Excitement:**

3: Ambivalent: It has merits (e.g., it reports state-of-the-art results, the idea is nice), but there are key weaknesses (e.g., it describes incremental work), and it can significantly benefit from another round of revision. However, I won't object to accepting it if my co-reviewers champion it.

**Missing References:**

For contrastive learning:

Feng Nan, Ramesh Nallapati, Zhiguo Wang, Cicero Nogueira dos Santos, Henghui Zhu, Dejiao Zhang, Kathleen McKeown, and Bing Xiang. 2021. Entity-level Factual Consistency of Abstractive Text Summarization. In Proceedings of the 16th Conference of the European Chapter of the Association for Computational Linguistics: Main Volume, pages 2727–2733, Online. Association for Computational Linguistics.

Griffin Adams, Bichlien Nguyen, Jake Smith, Yingce Xia, Shufang Xie, Anna Ostropolets, Budhaditya Deb, Yuan-Jyue Chen, Tristan Naumann, and Noémie Elhadad. 2023. What are the Desired Characteristics of Calibration Sets? Identifying Correlates on Long Form Scientific Summarization. In Proceedings of the 61st Annual Meeting of the Association for Computational Linguistics (Volume 1: Long Papers), pages 10520–10542, Toronto, Canada. Association for Computational Linguistics.

For clinical text evaluation:

Adams, Griffin, Jason Zucker, and Noémie Elhadad. "A meta-evaluation of faithfulness metrics for long-form hospital-course summarization." arXiv preprint arXiv:2303.03948 (2023).

For modeling:

Liu, Fenglin, et al. "Retrieve, reason, and refine: Generating accurate and faithful patient instructions." Advances in Neural Information Processing Systems 35 (2022): 18864-18877.

**Paper Topic And Main Contributions:**

They add two objectives to fine-tuning on medical summarization models.  The first is a contrastive objective (which is based on the CLIFF objective) but they use different methods for generating negative and positive contrast sets. The second is a novel objective (MKI) which encourages the model to generate medical concepts in the references.

They show improved performance over baseline models (BART, T5, mT5, and PEGASUS) when fine-tuning with their multi-task objectives.

**Questions For The Authors:**

- How is ensembling performed?
- Are the logits before softmax averaged across all reference tokens or is the MKI dot product computed at the token-level?

**Reasons To Accept:**

- The paper is very clearly and well-written. It is nicely polished with relevant related work, good diagrams, and thorough experimental analysis.

- The decision to not always include a reference as a positive example is smart and an easy win. That is a simple yet clear improvement over the perturbations in CLIFF. The other modifications are decided based on a qualitative error analysis, which is the right way to go about it.

- They test their methods on many downstream datasets and conduct a human evaluation of outputs.

**Reasons To Reject:**

- The experimental design doesn't adequately test the benefits of FaMeSumm over other CL methods. CLIFF is only tested against automatic metrics for one dataset (HQS). While it is compared with CLIFF on human annotations for 2 datasets (HQS and MDS), the results seem marginal.  I think it would help to run a pair-wise comparison annotation between CLIFF and FaMeSumm to see if it gets stronger results. It would also be helpful to directly compare FaMeSumm CL against CLIFF without the MKI loss, as it's hard to suss out what's diving the performance increase. For MKI, it's not very well motivated or compared to other methods which aim to boost the generation of salient content.  I would want to see a simpler version which just places a higher CE weight on medical terms.  It would be nice to also see an ablation of all the types of corruptions made to positive and negative sets.

- The MKI objective could be better motivated and explained. See questions below.

**Reproducibility:**

4: Could mostly reproduce the results, but there may be some variation because of sample variance or minor variations in their interpretation of the protocol or method.

**Reviewer Confidence:**

5: Positive that my evaluation is correct. I read the paper very carefully and I am very familiar with related work.

---

> ### Author Rebuttal · Authors · 2023-08-29
>
> We thank the reviewer for the helpful feedback!
>
>   To clarify the confusion about weakness 1 and 2, we organize our response into the following 4 points:
>
> **1. Comparison between our CL and CLIFF**
>
> We choose HQS to compare FameSumm with CLIFF because: (1) On MDS, it is hard to accurately implement CLIFF contrastive sets because it is a Chinese dataset. We would like to humbly point out that we compared CLIFF with human evaluation only on HQS, not MDS. (2) On RRS, our focus is to compare with other state-of-the-art models.
>
>   As shown in Table 7, our pair-wise human evaluation between CLIFF and FameSumm on HQS shows that the improvement of FameSumm is significant with 6% to 7% increase in Faithfulness rating.
>
>   To sort out the factors that drive performance increase, we report new experiment results by training FaMeSumm CL without MKI on both HQS and MDS datasets, and the performance is reported in the first two tables below (the first table is about HQS, and the second one is about MDS). Combining with the scores reported in the original paper, on HQS, we see that FaMeSumm CL has higher scores than CLIFF on all faithfulness metrics, which demonstrates the advantage of our proposed CL method. Its ROUGE and BERTScore are lower than CLIFF, because we disvalue many reference summaries by placing them into the negative sets. Paying the price of this trivial performance decrease, we obtain a consistent increase on faithfulness measures. Since FaMeSumm CL is not good at improving entity-based faithfulness, we see a marginal increase from FaMeSumm CL over CLIFF on FaR and C F1. We add MKI here to gain a more significant increase. In conclusion, FaMeSumm CL is better than CLIFF.
>
> **2. Complementary roles of CL and MKI**
>
>   Analyzing the second table, we aim to analyze the complementary roles of CL and MKI. We can see that FaMeSumm CL has better RL and BERTScore than MKI only, which shows that improving faithfulness from question answering and entailment aspects has the potential of improving general summarization quality. It also has better scores than plain mT5.
>
>  Contrary to FaMeSumm CL, MKI is most effective on entity-based faithfulness metrics as shown in the second table. It yields the highest C F1 and the second highest FaR. We can see that MKI has the strongest impact on C F1 and FaR.
>
>   We also trained the simpler version as suggested in the review. Technically speaking, this “simpler” version does not differ much from our MKI in terms of implementation difficulty and training efficiency (e.g., GPU memory consumption and training time used for 1 epoch). The reason is that the PyTorch implementation of cross entropy loss requires us to construct a weight vector that has the same dimension as the vocabulary of the backbone model (see our answer to question 2; these 2 implementations are very similar). Then, we compare their performance in the second table and find that MKI is significantly better. We offer two reasons to explain this phenomenon: (1) our MKI considers modeling the context tokens (2 tokens before medical terms and negative unigrams) as our unique contribution to this domain, and this greatly facilitates the generation of medical terms, since almost all existing language models generate tokens in an autoregressive fashion so modeling context tokens helps; (2) optimizing logits instead of weighted cross entropy loss has a more direct impact on the decoding process of language models.
>
> **3. Ablation of different types of contrastive sets**
>
>  In this third table, it is clear to see the reason why we disvalue some reference summaries for positive sets construction as mentioned in footnote 1. The last row dominates all faithfulness metrics while having slightly lower ROUGE and BERTScore. Since this paper mainly deals with improving faithfulness and the decrease is trivial, we place some reference summaries into negative sets if they fail our check described in Section 3.1.
>
> |                                              | QuestEval  | FaR        | SummaC     | C F1      | R1        | R2        | RL        | BERTScore  |
> |----------------------------------------------|------------|------------|------------|-----------|-----------|-----------|-----------|------------|
> | PEGASUS                                      | 0.3069     | 0.3188     | 0.4279     | 26.24     | 30.19     | 11.93     | 28.52     | 0.7427     |
> | PEGASUS + CLIFF                              | 0.3145     | 0.3217     | 0.4225     | 30.18     | 29.92     | 11.40     | 27.67     | 0.7401     |
> | PEGASUS + FaMeSumm                           | 0.3134     | **0.3492** | **0.4401** | **30.92** | **30.84** | **12.19** | **28.99** | **0.7456** |
> | FaMeSumm CL only (**new experiment**)            | **0.3157**     | 0.3222     | 0.4355     | 30.20     | 29.31     | 10.82     | 27.39     | 0.7352     |
>
> |                                                       | FaR    | C F1  | RL    | BERTScore |
> |-------------------------------------------------------|--------|-------|-------|-----------|
> | mT5*                                                  | 0.3784 | 36.96 | 32.43 | 0.7505    |
> | +CL* (**new experiment**)                                 | 0.4040 | 37.85 | 32.73 | 0.7518    |
> | +MKI*                                                 | 0.4072 | **39.30** | 32.27 | 0.7492    |
> | +FaMeSumm*                                            | **0.4177** | 38.57 | **33.05** | **0.7520**    |
> | Increase CE weights on medical terms* (**new experiment**) | 0.4024 | 37.79 | 31.73 | 0.7485    |
>
>
> |                                                                            | QuestEval  | FaR        | SummaC     | C F1      | R1        | R2        | RL        | BERTScore  |
> |----------------------------------------------------------------------------|------------|------------|------------|-----------|-----------|-----------|-----------|------------|
> | PEGASUS + FaMeSumm (all references are placed into positive sets as CLIFF) | 0.3105     | 0.3267     | 0.4138     | 28.90     | **31.14** | **12.29** | **29.09** | **0.7489** |
> | PEGASUS + FaMeSumm                                                         | **0.3134** | **0.3492** | **0.4401** | **30.92** | 30.84     | 12.19     | 28.99     | 0.7456     |
>
> **4. Motivation, explanation, and empirical evidence of MKI**
>
> The motivation of MKI is to facilitate the generation of summaries with more accurate entities, terms, and their logical relationships such as negation captured by context words, thus improving faithfulness in the medical domain. To this end, we design a loss term L_MKI to encourage models to maximize the prediction scores of the medical terms as well as their context words. Specifically, in Equation 1, we use the dot product between b_m and p which have the same dimension as the vocabulary size. Here, b_m records the frequency of medical terms and their context tokens, while p is the logits of medical terms and their context by averaging across all reference tokens. Our motivation has been further supported by our empirical results in Table 5, Table7, and new experiments above, demonstrating that MKI is most effective on entity-based faithfulness metrics.
>
> Q1: Thanks for your question! We provide more ensembling details here. On HQS benchmark, following the previous state-of-the-art (He et al., 2021), we rerank 4 system outputs (FaMeSumm fine-tuned with BioBART, T5, PEGASUS, and BART) based on four features: fidelity, length, consensus, and wellformedness. Computing a weighted sum of the four features, we pick the best summary from the 4 outputs for every test instance. Note that the previous state-of-the-art trained 4 systems on the union of training and validation data and utilized some ad-hoc techniques such as misspelling correction (mentioned in line 445 and 448), but **we do not need those to reach better performance**. The reason is that each of our single systems has better performance as shown in Table 3.
>
> On RRS-Indiana, the previous state-of-the-art (Dai et al., 2021) fine-tuned **16** PEGASUS (with different seeds) on the union of training and validation data and performed ensembling based on mutual similarity scores among any two system outputs. In contrast, we leverage the observations conducted on the training set in that paper and rerank system output out of **7** FaMeSumm (fine-tuned based on PEGASUS with different seeds) outputs.
>
> On RRS-Stanford, the previous state-of-the-art (Dai et al., 2021) fine-tuned **16** PEGASUS (with different seeds) on the training set with output normalization (this normalization is based on an observation mentioned in the previous paragraph). We reach better performance by **just using one** FaMeSumm (fine-tuned based on PEGASUS with different seeds) output. This FaMeSumm model is the same model as the last row of Table 4, except that length penalty parameter is additionally tuned on the validation set in order to encourage our model to generate longer output.
>
> We will describe these implementations with more details in our final version and share our ensembling code for each dataset.
>
> Q2: Thanks for your question! In Equation 1, both b_m and p have the same dimension as the vocabulary size. b_m records the frequency of medical terms and their context tokens, while p is the logits of medical terms and their context by averaging across all reference tokens. Therefore, the MKI dot product in Equation 1 is computed at vocabulary-level, not token-level. We will clarify this in our final version.
>
>
>
> Thanks a lot for bringing up these missing references. All these publications have related but different focuses than our work. We will add them into Section 6 in our final version.

---

### Official Review · Reviewer_Ptjw · 2023-08-07

**Soundness:** 4

**Excitement:**

4: Strong: This paper deepens the understanding of some phenomenon or lowers the barriers to an existing research direction.

**Paper Topic And Main Contributions:**

This work reports an NLP Engineering experiment for improving medical summarization systems (and reproduces several baselines) via training with loss functions targeted to increase faithfulness (against constructed negative samples) and topicality with respect to the inputs (as measured by weighted term mentions). It includes several model variants (based in PEGASUS, BART, T5), a comparison against GPT-3, an ablation over the objective modifications, several automated metrics (QuestEval, FaR, SummaC, C(oncept) F1, and ROUGE variants), and a human evaluation. The automated metrics show a possibly meaningful difference in SummaC and C F1, while all other automated metrics show only small (mostly marginally improved) results. The human metrics tell a more interesting story, with a clear improvement from adding the new objectives and a lack of general utility from many of the automated metrics.

**Questions For The Authors:**

(A) Are the backtranslated/data augmentation instances also verified for factuality?

(B) What is the final summary representation? Several works describe options from an average over tokens, an average over named entity tokens, or using the last token. Line 254 “last layer of the decoder from a language model” is not a clear description - that could be the last token or an average of all tokens.

(C) Why  are only a limited number of models evaluated for Tables 4 -8 (as opposed to all the models mentioned/evaluated in this work)?

(D) How was the hyperparameter search  conducted? Grid? Some other optimization? The appendix only specifies the objective (a weighted average of Rouge-2 and length ratio), but not the actual method of search.

(E) Similarly, is this tuning (particularly for the lambda weights) performed separately for the  ablations over loss functions? Repeating all parameter selection steps (particularly for the loss weightings, perhaps less so for learning rates and other hyperparameters)is required to answer the implicit question in ablation of “what if we only used  this particular setting”..

(F) What was the Fleiss Kappa on the reference?

(G) Table 6: what is the FAMESUMM ensemble over? My best guess is “all models” but not all models always have results shown.

(H) Are there any meaningful qualitative differences between summaries from the different models/configurations? As the automated metrics are (mostly) not capturing large differences between models and configurations, a qualitative evaluation of the actual changes they have on model text would be useful.

(I) What are the GPT-3 ROUGE scores?

**Reasons To Accept:**

The paper runs a reasonable set of experiments, with methods not previously applied as thoroughly or as well to this domain, is mostly clear on the experimental details, and shows improvements on a human evaluation.

**Reasons To Reject:**

(A) Table 5 - Averaging scores over three random seeds is not a fair evaluation. The paper must specify how a single answer is chosen, whereas the average conflates aggregates over three scores.

(B) The paper leaves many open questions, some of which might be addressable via some editing, appendix entries, and/or automated methods (questions (A), (B), (C), (D), (E), (G), (I)), while others might require more experimental work ((F), (H)).

(C) Question (H), regarding finding any qualitative differences, is an important question for understanding what the methods in this work actually do.

**Reproducibility:**

4: Could mostly reproduce the results, but there may be some variation because of sample variance or minor variations in their interpretation of the protocol or method.

**Reviewer Confidence:**

3: Pretty sure, but there's a chance I missed something. Although I have a good feel for this area in general, I did not carefully check the paper's details, e.g., the math, experimental design, or novelty.

**Typos Grammar Style And Presentation Improvements:**

(A) I do not read any Chinese - since it seems the authors are bilingual , it may be beneficial for readers to include a translation of the examples in the Appendix. I have not counted this against the paper.

(B) RQ1 (line 377) - this should be “Does FAMESUMM” not “How can” as the how of the method causing improvements is speculative. Rephrasing this makes it clear that RQ1 and RQ3 are not actually different.

---

> ### Author Rebuttal · Authors · 2023-08-29
>
> We thank the reviewer for the helpful feedback!
>
> Weakness (A): The reason that we used averaged scores on transformer-based models is to try to be fair for all models in Table 5. Based on our observation and a recent work (Mosbach et al., 2021), we noticed that the stability of transformer-based models is significantly lower than the baseline based on the pointer-generator network (Joshi et al., 2020). Since we do not want to report “lucky” (or “unlucky”) scores for our models, we report averaged scores in Table 5. Moreover, we did not choose a single score out of 3. Instead, we compute an averaged score for every metric in Table 5.
>
> Weakness (B) and (C): We address them in detail through the following questions.
>
> Q(A): When we constructed our positive sets, we had manually verified at least 20 examples for each dataset through random sampling, and we did not notice any unfaithful errors. Considering the fact CLIFF also utilized this method, we think the translated instances generally provide high-quality positive examples. As there is no theoretical guarantee on their faithfulness, we rank this method as the last method (discussed in line 208) if we are still behind the minimum size requirement of positive sets (there has to be at least 2 positive examples). We will stop the construction of positive sets whenever we meet this requirement, so most training instances do not have this kind of translated data.
>
> Q(B): Thanks for pointing out this ambiguity! The outputs produced by the last layer of the decoder are averaged over all tokens as the final summary representation.
>
> Q(C): Our experiments are designed to answer our research questions as listed in the beginning of Section 5, and we believe the current set of experiments are sufficient to reach the claimed contribution. In addition, we will add more experimental results in our final version as shown in the response to Reviewer 1 and Reviewer 3.
>
> Q(D): Yes, we performed grid search on the validation set of each dataset.
>
> Q(E): Yes, we performed tuning on those lambda weights separately. All lambda values can be viewed in Appendix C.
>
> Q(F): Thanks for your question. Table 7 aims to evaluate different models’ performance by doctors, so we did not ask doctors to evaluate reference summaries, and we do not have the Fleiss Kappa on reference. Our human evaluation is purely conducted by doctors. It is comprehensive by covering two datasets and five baselines while showing substantial improvements in faithfulness ratings (e.g., 6% to 7% improvement over CLIFF on HQS).
>
> Q(G): Thanks for your question! We provide more ensembling details here. On HQS benchmark, following the previous state-of-the-art (He et al., 2021), we rerank 4 system outputs (FaMeSumm fine-tuned with BioBART, T5, PEGASUS, and BART) based on four features: fidelity, length, consensus, and wellformedness. Computing a weighted sum of the four features, we pick the best summary from the 4 outputs for every test instance. Note that the previous state-of-the-art trained 4 systems on the union of training and validation data and utilized some ad-hoc techniques such as misspelling correction (mentioned in line 445 and 448), but **we do not need those to reach better performance**. The reason is that each of our single systems has better performance as shown in Table 3.
>
> On RRS-Indiana, the previous state-of-the-art (Dai et al., 2021) fine-tuned **16** PEGASUS (with different seeds) on the union of training and validation data and performed ensembling based on mutual similarity scores among any two system outputs. In contrast, we leverage the observations conducted on the training set in that paper and rerank system output out of **7** FaMeSumm (fine-tuned based on PEGASUS with different seeds) outputs.
>
> On RRS-Stanford, the previous state-of-the-art (Dai et al., 2021) fine-tuned **16** PEGASUS (with different seeds) on the training set with output normalization (this normalization is based on an observation mentioned in the previous paragraph). We reach better performance by **just using one** FaMeSumm (fine-tuned based on PEGASUS with different seeds) output. This FaMeSumm model is the same model as the last row of Table 4, except that length penalty parameter is additionally tuned on the validation set in order to encourage our model to generate longer output.
>
>
> Q(H): For qualitative analysis, Figure 2 and Section 5.5 present a case study (translation provided) to showcase the effectiveness of FaMeSumm compared with baselines such as mT5 and PEGASUS. Our takeaway message about this case study is that FaMeSumm can generate more accurate domain-specific terms, entities, and their relationships such as negation to reduce popular unfaithful errors mentioned in Table 1, which improves summarization faithfulness. We will add more examples in the Appendix in our final version.
>
> Q(I): We just computed the ROUGE scores for GPT outputs, and the R1, R2, and RL of GPT-3 on HQS are 30.51, 11.61, and 27.85. These scores are lower than FaMeSumm in Table 3. We will add these in our final version.
>
> Typo (A): Thanks for pointing this out. We have translation in Figure 2,and we will surely add translation for Table 1 as well.
>
> Typo (B): Good point! We will rephrase line 377 to “Does FaMeSumm…”.

---

### Official Review · Reviewer_BQfn · 2023-08-11

**Soundness:** 3

**Excitement:**

3: Ambivalent: It has merits (e.g., it reports state-of-the-art results, the idea is nice), but there are key weaknesses (e.g., it describes incremental work), and it can significantly benefit from another round of revision. However, I won't object to accepting it if my co-reviewers champion it.

**Missing References:**

This would be a stronger baseline:
* Ming Zhang, Shuai Dou, Ziyang Wang, and Yunfang Wu. 2022. Focus-Driven Contrastive Learning for Medical Question Summarization. In Proceedings of the 29th International Conference on Computational Linguistics, pages 6176–6186, Gyeongju, Republic of Korea. International Committee on Computational Linguistics.

More work on contrastive learning for summarization (though not necessarily medical summarization):
* Zhichao Geng, Ming Zhong, Zhangyue Yin, Xipeng Qiu, and Xuanjing Huang. 2022. Improving Abstractive Dialogue Summarization with Speaker-Aware Supervised Contrastive Learning. In Proceedings of the 29th International Conference on Computational Linguistics, pages 6540–6546, Gyeongju, Republic of Korea. International Committee on Computational Linguistics.
* Yixin Liu and Pengfei Liu. 2021. SimCLS: A Simple Framework for Contrastive Learning of Abstractive Summarization. In Proceedings of the 59th Annual Meeting of the Association for Computational Linguistics and the 11th International Joint Conference on Natural Language Processing (Volume 2: Short Papers), pages 1065–1072, Online. Association for Computational Linguistics.
* Jinpeng Hu, Zhuo Li, Zhihong Chen, Zhen Li, Xiang Wan, and Tsung-Hui Chang. 2022. Graph Enhanced Contrastive Learning for Radiology Findings Summarization. In Proceedings of the 60th Annual Meeting of the Association for Computational Linguistics (Volume 1: Long Papers), pages 4677–4688, Dublin, Ireland. Association for Computational Linguistics.
* Chenxin An, Ming Zhong, Zhiyong Wu, Qin Zhu, Xuanjing Huang, and Xipeng Qiu. 2022. CoLo: A Contrastive Learning Based Re-ranking Framework for One-Stage Summarization. In Proceedings of the 29th International Conference on Computational Linguistics, pages 5783–5793, Gyeongju, Republic of Korea. International Committee on Computational Linguistics.

Related to the idea of permuting attribute values to get negative examples (line 234):
* Zheng Zhao, Shay B. Cohen, and Bonnie Webber. 2020. Reducing Quantity Hallucinations in Abstractive Summarization. In Findings of the Association for Computational Linguistics: EMNLP 2020, pages 2237–2249, Online. Association for Computational Linguistics.

Taxonomizing factual errors in summarization models:
* Artidoro Pagnoni, Vidhisha Balachandran, and Yulia Tsvetkov. 2021. Understanding Factuality in Abstractive Summarization with FRANK: A Benchmark for Factuality Metrics. In Proceedings of the 2021 Conference of the North American Chapter of the Association for Computational Linguistics: Human Language Technologies, pages 4812–4829, Online. Association for Computational Linguistics.

**Paper Topic And Main Contributions:**

This work proposes a customization of contrastive learning for medical summarization. They design an alternate strategy for constructing positive and negative examples for contrastive learning and introduce an additional loss that rewards the generation of medical terms from the input. They conduct automatic and manual evaluation of models trained with their three-part loss, and find that these models have higher factuality than models trained with only cross-entropy loss or a different contrastive learning setup.

**Questions For The Authors:**

A. Is L_CL equivalent to CLIFF with a different selection strategy for positive/negative examples?

B. For table 5, what is the relative performance of CLIFF? What about using only your L_CL?

C. You describe FaMeSumm as "cost-efficient" (line 105) because only one model must be trained, but also state that it requires 48 GB of memory to finetune PEGASUS with batch size 2 (in Limitations). It's not clear to me that training 1 big model with contrastive learning is more efficient than prior approaches (finetuning without contrastive learning and training a second, smaller model as a faithfulness detector). Can you quantify the difference in cost (memory or GPU-hours) between your method and prior work?

D. Your results in Table 6 are generally very small increases in ROUGE. Are these results from single runs or averaged over multiple runs? And did you perform statistical testing?

E. The validation of reference summaries removes summaries with medical terms not mentioned in the input, but the datasets include health question summarization and dialogue summarization. The patients posing questions/speaking to the doctor may not use the correct medical terms, but that does not make the reference that uses those terms unfaithful. Did you notice any decrease in performance over test set examples that require the model to generate medical terms not found in the input? If so, do you consider this to be a positive or negative impact of the method?

**Reasons To Accept:**

S1. The observation in footnote 1 (that contrastive learning is less effective when the positive examples include gold reference summaries that use medical terms not found in the input) is surprising, and a useful insight. (Though the authors do note that a similar issue is discussed in Adams (2022).).

S2. The additional loss term L_MKI appears novel and has a clear intuitive justification. The same underlying concept of upweighting relevant domain terms during training could be useful across domains.

S3. The authors perform a detailed human evaluation, and in this evaluation, the method improves factuality over CLIFF.

**Reasons To Reject:**

W1. The ablations of the role of each loss term are incomplete. In particular, this work introduces two ideas: a different strategy for selecting/excluding examples for contrastive learning and an additional loss that rewards generating medical terms from the input. However, the new selection strategy is not compared to prior selection strategies (CLIFF) on its own, and so it is impossible to disentangle whether the improvement over prior work comes from the selection strategy or the additional loss term L_MKI.

W2. Zhang et al. 2022 would be a stronger baseline than CLIFF, as it is also contrastive learning approach focused on medical summarization:
>Ming Zhang, Shuai Dou, Ziyang Wang, and Yunfang Wu. 2022. Focus-Driven Contrastive Learning for Medical Question Summarization. In Proceedings of the 29th International Conference on Computational Linguistics, pages 6176–6186, Gyeongju, Republic of Korea. International Committee on Computational Linguistics.

W3. In several places, the authors indicate that "only a few papers" (line 49, 552) focus on faithfulness in medical summarization, but there is ample prior work in this area. I feel the authors are slightly misrepresenting the state of the research in this direction. Additionally, the paper would benefit from an additional section in the related work discussing the prior work on contrastive learning for summarization.

W4. The taxonomy of error categories in Table 1 is similar to the taxonomy proposed in Pagnoni et al 2021, and would benefit from comparing/engaging with that taxonomy:

> Artidoro Pagnoni, Vidhisha Balachandran, and Yulia Tsvetkov. 2021. Understanding Factuality in Abstractive Summarization with FRANK: A Benchmark for Factuality Metrics. In Proceedings of the 2021 Conference of the North American Chapter of the Association for Computational Linguistics: Human Language Technologies, pages 4812–4829, Online. Association for Computational Linguistics.

**Edit after rebuttal**: The authors have run additional experiments to address W1 and W2, and I have updated my soundness score accordingly (2->3).

**Reproducibility:**

5: Could easily reproduce the results.

**Reviewer Confidence:**

4: Quite sure. I tried to check the important points carefully. It's unlikely, though conceivable, that I missed something that should affect my ratings.

**Typos Grammar Style And Presentation Improvements:**

line 54-56:
> as many summarization approaches are based on language models that are pretrained on general domain text, they are inadequate

As I understand it, your method is also based on pretrained models (with additional loss terms customized for the domain). This phrasing suggests to me that your method does not use models pretrained on general domain text.

Section 4.4 could be moved earlier in the paper-- around line 268-270, I was wondering how you performed the identification of medical terms. It would be nice to have that information in the first place where you describe using medical terms.

Footnote 1 is interesting, and further elaboration on this point would be appreciated.

---

> ### Author Rebuttal · Authors · 2023-08-29
>
> We thank the reviewer for the helpful feedback!
>
> Weakness1
>
>   To sort out the factors that drive performance increase, we report new experiment results by training FaMeSumm CL without MKI on both HQS and MDS datasets, and the performance is reported in the first two tables below (the first table is about HQS, and the second one is about MDS). Combining with the scores reported in the original paper, on HQS, we see that FaMeSumm CL has higher scores than CLIFF on all faithfulness metrics, which demonstrates the advantage of our proposed CL method. Its ROUGE and BERTScore are lower than CLIFF, because we disvalue many reference summaries by placing them into the negative sets. Paying the price of this trivial performance decrease, we obtain a consistent increase on faithfulness measures. Since FaMeSumm CL is not good at improving entity-based faithfulness, we see a marginal increase from FaMeSumm CL over CLIFF on FaR and C F1. We add MKI here to gain a more significant increase. In conclusion, FaMeSumm CL is better than CLIFF.
>
>   Analyzing the second table, we aim to analyze the complementary roles of CL and MKI. We can see that FaMeSumm CL has better RL and BERTScore than MKI only, which shows that improving faithfulness from question answering and entailment aspects has the potential of improving general summarization quality. It also has better scores than plain mT5.
>
>   Contrary to FaMeSumm CL, MKI is most effective on entity-based faithfulness metrics as shown in the second table. It yields the highest C F1 and the second highest FaR. We can see that MKI has the strongest impact on C F1 and FaR.
>
>
> |                                              | QuestEval  | FaR        | SummaC     | C F1      | R1        | R2        | RL        | BERTScore  |
> |----------------------------------------------|------------|------------|------------|-----------|-----------|-----------|-----------|------------|
> | PEGASUS                                      | 0.3069     | 0.3188     | 0.4279     | 26.24     | 30.19     | 11.93     | 28.52     | 0.7427     |
> | PEGASUS + CLIFF                              | 0.3145     | 0.3217     | 0.4225     | 30.18     | 29.92     | 11.40     | 27.67     | 0.7401     |
> | PEGASUS + FaMeSumm                           | 0.3134     | **0.3492** | **0.4401** | **30.92** | **30.84** | **12.19** | **28.99** | **0.7456** |
> | PEGASUS + Zhang et al. 2022 (**new experiment**) | **0.3193** | 0.3042     | 0.4322     | 28.00     | 29.27     | 11.01     | 27.32     | 0.7396     |
> | FaMeSumm CL only (**new experiment**)            | 0.3157     | 0.3222     | 0.4355     | 30.20     | 29.31     | 10.82     | 27.39     | 0.7352     |
>
>
>
>
> |                                    | FaR        | C F1      | RL        | BERTScore  |
> |------------------------------------|------------|-----------|-----------|------------|
> | mT5*                               | 0.3784     | 36.96     | 32.43     | 0.7505     |
> | FaMeSumm CL only* (**new experiment**) | 0.4040     | 37.85     | 32.73     | 0.7518     |
> | +MKI*                              | 0.4072     | **39.30** | 32.27     | 0.7492     |
> | +FaMeSumm*                         | **0.4177** | 38.57     | **33.05** | **0.7520** |
>
>
> Weakness 2: We appreciate this insight! In the first table, we report the performance of Zhang et al. 2022. This paper claims that it is different from CLIFF in terms of negative sample construction method, so it shares the same positive sets as CLIFF. It indeed leads CLIFF on QuestEval and SummaC, which means that Zhang et al. 2022 is a stronger baseline in terms of question answering and entailment aspects of faithfulness. The reason behind is likely due to its manipulation on question focus for the medical question summarization task. Our FaMeSumm CL still outperforms Zhang et al. 2022 on most metrics. Except QuestEval, combining MKI with FaMeSumm CL provides us a greater performance boost. Note that FaMeSumm is designed to handle various summarization tasks in healthcare while Zhang et al. 2022 has a narrower scope and only conducted experiments on question summarization.
>
> Weakness 3: We apologize for the misunderstanding! We did not mean to ignore the ample amount of prior work. In line 49, we meant to say that not many works **“investigated”** faithfulness by providing systematic analysis and categorization of faithfulness errors on medical domain as what we did in Section 2. In line 552, we meant to say that not all works **“directly modeled”** faithfulness by adding inductive bias informed by faithfulness through model optimization, as many papers improve faithfulness through modifying training data or other effective means. We are more interested in those papers that directly modeled faithfulness during training, since they are the direct competitor of our work. We will clarify these points about prior work in our final version.
>
> Weakness 4: Thanks for pointing this out! Yes, the taxonomy proposed in Pagnoni et al. 2021 is similar to ours. We both have some same error types such as entity error. While Pagnoni et al. 2021 is on general domain, we propose new error types that are specific to the medical domain. For example, low specificity error in our taxonomy is particularly important for the medical domain, since ignoring certain specifiers can be detrimental in healthcare. We recommend combining all kinds of taxonomies in order to comprehensively investigate unfaithful errors.
>
> Q (A): Yes. We provided a direct comparison between these two as discussed above.
>
> Q (B): It is hard to accurately implement CLIFF contrastive sets on MDS because it is a Chinese dataset. For example, CLIFF utilized machine translation for positive set construction (same technique as the one in line 215), but we did not find a comparable translation tool that works at the similar level of performance on Chinese data. Note that we do not need to apply machine translation on MDS as a part of FaMeSumm, since we could already meet the minimum size requirement of the positive set by following the prior data augmentation techniques discussed in lines from 201 to 217. As for using only our L_CL, its performance is discussed above for weakness 1.
>
> Q (C): Thanks for your question! We would like to clarify that the 48GB requirement mentioned in the Limitations section applied only when we used PEGASUS-large or BART-large. For all of our experiments reported in this paper that use PEGASUS-large or BART-large, we never needed to go beyond batch size of 4, so the most memory we possibly needed was 48*2=96GB. However, in Chen et al. 2022, they claimed that 4 NVIDIA A100 GPUs are needed for their experiments based on BART-large and pegasus-xsum. Although we are not sure whether each of their A100 has 40GB or 80GB, even the most optimistic estimate for them (or the most pessimistic estimate for us) is that FaMeSumm would save at least 40*4-96=64GB of GPU memory.
>
>
> Q (D): Thanks for your question! On HQS benchmark, following the previous state-of-the-art (He et al., 2021), we rerank 4 system outputs (FaMeSumm fine-tuned with BioBART, T5, PEGASUS, and BART) based on four features: fidelity, length, consensus, and wellformedness. Computing a weighted sum of the four features, we pick the best summary from the 4 outputs for every test instance. Note that the previous state-of-the-art trained 4 systems on the union of training and validation data and utilized some ad-hoc techniques such as misspelling correction (mentioned in line 445 and 448), but **we do not need those to reach better performance**. The reason is that each of our single systems has better performance as shown in Table 3.
>
> On RRS-Indiana, the previous state-of-the-art (Dai et al., 2021) fine-tuned **16** PEGASUS (with different seeds) on the union of training and validation data and performed ensembling based on mutual similarity scores among any two system outputs. In contrast, we leverage the observations conducted on the training set in that paper and rerank system output out of **7** FaMeSumm (fine-tuned based on PEGASUS with different seeds) outputs.
>
> On RRS-Stanford, the previous state-of-the-art (Dai et al., 2021) fine-tuned **16** PEGASUS (with different seeds) on the training set with output normalization (this normalization is based on an observation mentioned in the previous paragraph). We reach better performance by **just using one** FaMeSumm (fine-tuned based on PEGASUS with different seeds) output. This FaMeSumm model is the same model as the last row of Table 4, except that length penalty parameter is additionally tuned on the validation set in order to encourage our model to generate longer output.
>
> We do not aim to make a huge improvement on ROUGE in Table 6. Instead, we achieved better scores with fewer base models and ad-hoc techniques, since each of our base models (presented in Table 3 and 4) has better performance than its baseline. We will describe these implementations with more details in our final version and share our ensembling code for each dataset.
>
> Q (E): We appreciate this insight! As mentioned in our footnote, our decision of validating and removing reference summaries with novel medical terms is based on our initial experiments about analyzing whether to place all references into positive sets. To give more details, we provide a third table below on experiments on the HQS dataset. As you pointed out, devaluing some references (placing them into the negative sets when they have novel medical terms) will hurt general summarization quality to a small extent with a performance decrease on ROUGE and BERTScore. However, we see a consistent and a more significant improvement on faithfulness scores as shown below. Paying the price of a trivial performance decrease on ROUGE and BERTScore, we view our validation of reference summaries as a contribution to related work. When patients do not use the correct medical terms, our method can be further extended to incorporate these examples that require the model to generate medical terms not in the input, and we will leave this as future work.
>
> |                                                                            | QuestEval  | FaR        | SummaC     | C F1      | R1        | R2        | RL        | BERTScore  |
> |----------------------------------------------------------------------------|------------|------------|------------|-----------|-----------|-----------|-----------|------------|
> | PEGASUS + FaMeSumm (all references are placed into positive sets as CLIFF) | 0.3105     | 0.3267     | 0.4138     | 28.90     | **31.14** | **12.29** | **29.09** | **0.7489** |
> | PEGASUS + FaMeSumm                                                         | **0.3134** | **0.3492** | **0.4401** | **30.92** | 30.84     | 12.19     | 28.99     | 0.7456     |
>
> Regarding style and presentation improvement, we will move Section 4.4 earlier in our final version to eliminate the confusion and modify our phrasing in lines 54-56 to “as many summarization approaches do not add inductive bias informed by faithfulness through model optimization”. Further elaboration on footnote 1 is provided above in Q (E).
>
>
> Thanks a lot for bringing up those missing references. We will add them into Section 6 in our final version.

---

### Meta-Review · Area_Chair_RTMJ · 2023-09-19

**Recommendation:** 4

**Metareview:**

The paper explores methods for improving the faithfulness of medical summarization. It offers a solution to a tangible issue in summarization systems. The paper is commendable for its clear writing, inclusion of related work, and comprehensive experimental analysis. However, reviewers also pointed out concerns, including an omission of prior work on contrastive learning for summarization, similarities between the error taxonomy in Table 1 and Pagnoni et al. 2021, insufficient testing of FaMeSumm's benefits compared to other CL methods, and several unanswered questions that may need further experimentation.

---

### Decision · Program_Chairs · 2023-10-07

**Decision:**

Accept-Main

**Comment:**

The paper explores methods for improving the faithfulness of medical summarization. It offers a solution to a tangible issue in summarization systems. The paper is commendable for its clear writing, inclusion of related work, and comprehensive experimental analysis. However, reviewers also pointed out concerns, including an omission of prior work on contrastive learning for summarization, similarities between the error taxonomy in Table 1 and Pagnoni et al. 2021, insufficient testing of FaMeSumm's benefits compared to other CL methods, and several unanswered questions that may need further experimentation.